# In Vivo Testing of a Second-Generation Prototype Accessory for Single Transapical Left Ventricular Assist Device Implantation

**DOI:** 10.3390/bioengineering11080848

**Published:** 2024-08-19

**Authors:** Florian Meissner, Michelle Costa Galbas, Hendrik Straky, Heiko Vestner, Manuela Schoen, Marius Schimmel, Johanna Reuter, Martin Buechsel, Johannes Dinkelaker, Heidi Cristina Schmitz, Martin Czerny, Wolfgang Bothe

**Affiliations:** 1Department of Cardiovascular Surgery, Medical Center—University of Freiburg, Faculty of Medicine, University of Freiburg, Hugstetter Strasse 55, 79106 Freiburg, Germany; 2Institute for Clinical Chemistry and Laboratory Medicine, Medical Center—University of Freiburg, Faculty of Medicine, University of Freiburg, Hugstetter Strasse 55, 79106 Freiburg, Germany; 3Center for Experimental Models and Transgenic Service, Medical Center—University of Freiburg, Faculty of Medicine, University of Freiburg, Stefan-Meier-Strasse 17, 79104 Freiburg, Germany

**Keywords:** heart failure, left ventricular assist device, medical device development, transventricular outflow graft, animal testing, pigs, in vivo

## Abstract

A new accessory was developed to allow implantation of left ventricular assist devices (LVADs) without requiring an anastomosis to the ascending aorta. The accessory combines the LVAD inflow and outflow into a dual-lumen device. Initial prototypes encountered reduced pump performance in vitro, but a second-generation prototype successfully addressed this issue. This feasibility study aimed to demonstrate the anatomic fit, safe implantation, and hemodynamic effectiveness of the LVAD with the accessory. The accessory was implanted in ten female pigs (104 ± 13 kg). Following sternotomy and apical coring under cardiopulmonary bypass, a balloon catheter was retrogradely inserted and exteriorized through the coring site, where it was inflated within the distal third of the outflow graft. It was utilized to pull the accessory’s outflow across the aortic valve. After LVAD attachment, the catheter was removed. Echocardiography revealed no relevant valve regurgitation post-implantation. During ramp testing, pump flow increased from 3.7 ± 1.2 to 5.4 ± 1.2 L/min. Necropsy confirmed correct accessory placement in nine animals. No valve lesions or device thrombosis were observed. The accessory enabled LVAD implantation without compromising pump performance. Future work includes design refinements for implantation without cardiopulmonary bypass and long-term testing in a chronic heart failure model.

## 1. Introduction

More than 64 million people worldwide suffer from heart failure (HF), causing a high social and economic burden [1]. The HF prevalence is increasing, currently ranging from 1 to 3%. As of 2023, each HF patient incurs an average cost of approximately USD 27,000 per year due to direct expenses, comorbidities, invasive procedures, medications, and diagnostic tests [2]. Left ventricular assist devices (LVADs) allow mechanical left ventricular (LV) unloading and increase cardiac output in patients with severe HF. These devices provide both short- and long-term mechanical circulatory support (MCS), serving as a bridge to transplant, aiding myocardial recovery or destination therapy in adult and pediatric HF patients with an appropriate body surface area [3].

While LVAD technology has undergone continuous advancements, its use is associated with hemocompatibility-associated complications, including thromboembolism, stroke, and gastrointestinal bleeding [4]. Introducing fully magnetically levitated centrifugal-flow LVADs, such as the HeartMate 3 (HM3, Abbott, Chicago, IL, USA), which features relatively wide blood-flow pathways and artificial pulsatility, represents a significant advancement. Compared to mechanical-bearing axial-flow LVADs, the HM3 has significantly improved the short and intermediate survival rates of LVAD patients, reducing the incidence of adverse events [5,6].

While conventional LVAD implantation is performed through a median sternotomy and under cardiopulmonary bypass (CPB), less-invasive approaches combine, for example, an upper hemi-sternotomy with an anterior lateral thoracotomy or use a bilateral thoracotomy approach, reducing the surgical trauma, blood loss, and transfusion requirements [7]. In the past, off-pump implantation techniques have been proposed, such as inserting the LVAD into the LV under rapid-pacing [8,9]. Combining a less-invasive approach and avoiding CPB has several advantages, such as shortening operating and recovery time [10,11].

So far, conventional LVADs, such as the HM3, require surgical access to the LV apex to insert the inflow cannula and the ascending aorta (AAo) for anastomosing the outflow graft (OG). Aiming for a single transapical approach, which was discussed in the 1960s and 1970s for rapid cannulation of patients with cardiogenic shock [12,13], multiple miniaturized VAD (MVAD) designs have been proposed [14,15,16]. HeartWare (Miami Lakes, FL, USA) developed an MVAD prototype for minimally invasive off-pump implantation through anterolateral mini-thoracotomy. The continuous axial-flow pump allowed sufficient perfusion during acute and chronic bovine testing [14]. A redesigned MVAD with improved anatomical fit and performance was successfully tested in human cadavers and acute and chronic bovine experiments [15]. However, the device was never approved for HF patients. In addition to durable LVAD systems, percutaneous temporary MCS devices such as the Impella (Abiomed, Danvers, MA, USA) have become essential for treating patients with cardiogenic shock. The application of these devices can be limited by issues related to vascular access and intravascular hemolysis caused by shear stress from axial pumping [17]. Patients’ mobility is also restricted after implantation. Alternatively, transapical LV cannulation with a dual-lumen cannula for venoarterial-extracorporeal membrane oxygenation (VA-ECMO) has been proposed [18,19,20]. A study involving 50 patients demonstrated its safety and efficacy, showing mean flow rates of more than 5 L/min and a 30-day survival greater than 50%. However, approximately one in six patients experienced postoperative bleeding complications, defined as transfusion of more than three units of packed red blood cells in one day or intervention [21].

Our group is developing an accessory for LVADs that could allow for less-invasive implantation through anterolateral mini-thoracotomy. The accessory combines the inflow and the OG in a dual-lumen device. The inflow guides the blood towards the pump at the LV apex, while the transventricular OG redirects it across the aortic valve (AV) into the AAo (Figure 1a). This approach eliminates the need for access to the AAo for OG anastomosis. Prospectively, LVAD implantation might be performed without CPB on the beating heart using a customized introducer sheath and coring tool.

The first-generation prototype has already been implanted in juvenile pigs [22]. However, in vitro testing using a mock circulatory loop revealed a significant pressure decrease as the pump flow rates increased (data not published). Therefore, the flow channels were optimized to maintain a constant channel cross-sectional area. Featuring an OG diameter of 12 instead of 8 mm, the flow-optimized second-generation prototype (Figure 1b,c) revealed significantly lower pressure loss in vitro than the first-generation. Nevertheless, in the same in vitro setting, it reduced the flow rate and pressure head compared to LVAD only [23]. Moreover, there were no significant differences between the second-generation accessory and LVAD only in terms of hemolysis and blood damage in vitro [24]. The current study aimed to evaluate the anatomic fit, implantation safety, and hemodynamic performance of the LVAD combined with the accessory in vivo.

**Figure 1 bioengineering-11-00848-f001:**
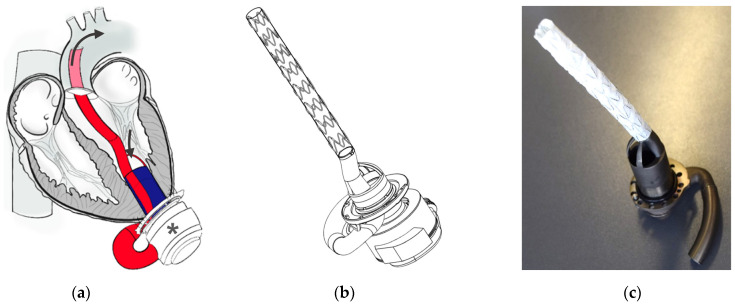
Second-generation accessory for left ventricular assist devices: (**a**) Schematic representation of the accessory attached to a pump (*). The inflow (blue) directs the blood from the left ventricle to the apical pump. The outflow (red) redirects it transventricularly across the aortic valve into the ascending aorta. Reprinted from “Impact of an Accessory for Left Ventricular Assist Devices on Device Flow and Pressure Head In Vitro” by Meissner et al. [23]. (**b**) Computer-aided design. (**c**) Titanium-made accessory with customized stent graft as outflow graft.

## 2. Materials and Methods

### 2.1. Device Design

The second-generation LVAD accessory was designed using *SolidWorks* (version 2021 SP3.0) [25] as computer-aided design software. It underwent in vitro and in silico characterization [23,24,26]. For in vivo testing, a 3D-printed titanium prototype was used (maximal diameter 28 mm, length for animal testing 40 mm, weight 70 g). The design comprises two flow channels for blood inflow and outflow. The inflow channel directs blood from the LV to the connected pump. A vascular graft or medical-grade tubing directs the blood further from the pump outlet to the accessory’s outflow channel. The outflow channel divides into two smaller channels as it enters the LV apex. These channels recombine within the LV and continue into a stent graft applied as transventricular OG (diameter of 12 mm). The stent graft, made of Dacron and nitinol springs, is positioned across the AV (Figure 1b). The prototype was combined with a centrifugal-flow HM3. A customized sewing ring made of titanium was used to attach the LVAD accessory to the epicardial surface of the LV apex.

### 2.2. Animal Testing

All experiments were approved by the local committee for animal experiments (Regierungspräsidium Freiburg, Germany, 35-9185.81/G-22/006). The research complied with the German animal protection law (TierSchG) and the ARRIVE guidelines [27,28]. Pigs were selected for this study due to their physiological and anatomical similarities to the human cardiovascular system [29]. After developing the implantation technique in three animals, the LVAD accessory was implanted in a standardized manner in ten juvenile German Landrace pigs. Before surgery, all animals were housed in the experimental animal facility of the University of Freiburg and under the care of experienced staff veterinarians. The pigs were provided with standard pellet chow (20 g/kg) and had access to water ad libitum. They were kept under controlled environmental conditions at 20 °C, 75 ± 5% humidity, and a 13/11 h light/dark cycle.

All animals underwent total intravenous anesthesia for surgery. For initial sedation, all animals received pre-medication with a combination of ketamine (20 mg/kg IM) and midazolam (0.5 mg/kg IM). Anesthesia was induced using propofol (2–4 mg/kg IV) and vecuronium (0.2 mg/kg IV). Following endotracheal intubation, anesthesia, and muscle relaxation were maintained with propofol (10–15 mg/kg/h IV), fentanyl (5–10 μg/kg/h IV), and vecuronium (0.2–0.4 mg/kg/h IV), respectively. Ringer’s solution (10 mL/kg/h IV) and noradrenaline (10–15 µg/kg/h IV) were administered to ensure fluid replacement and hemodynamic stability, respectively. Pressure-controlled ventilation was utilized, with the settings adjusted to maintain physiological conditions: FiO_2_ (20–40%), tidal volume (6–8 mL/kg), initial PEEP (5–8 mbar), and respiratory frequency (12–18/min).

Arterial blood pressure, central venous pressure (CVP), and pulmonary artery pressure (PAP) were monitored continuously. Intermittent cardiac output measurements were taken every 20 min using a pulmonary artery catheter with thermodilution. Blood flow in the left arterial carotid artery was measured at 20 min intervals using a perivascular transit-time flow probe (PV Probe 5 and 6 mm, Medistim ASA, Oslo, Norway). Pump data from the HM3 system monitor were recorded manually every minute.

Arterial blood samples were collected at four time points: (1) after arterial catheterization (baseline), (2) before CPB, (3) before, and (4) after LVAD support (before euthanasia). Blood cell count, blood chemistry, and plasma-free hemoglobin were measured at each time. Hemostatic markers, including von Willebrand factor (vWF) antigen and activity, were determined at points (1), (3), and (4). After sternotomy, epicardial and epiaortic echocardiography were performed using a CX50 point-of-care ultrasound system with an S4-2 transducer (Philips Healthcare, Hamburg, Germany). Apical, long-, and short-axis views were obtained following a standardized protocol [30]. LV function was assessed before and after device implantation, and color Doppler was used to examine valvular function. Moreover, the distance between the LV apex and sinotubular junction was measured to determine the minimum OG length. The study plan is summarized in Figure 2.

### 2.3. Surgical Procedure

All animals were placed in a supine position. A flexible guidewire was inserted either through the right common carotid or the left femoral artery. Subsequently, a pigtail catheter facilitated the exchange of the flexible guidewire with an extra-stiff one. A balloon catheter was then retrogradely delivered into the LV outflow tract. For CPB, the right femoral vein and artery were cannulated. Median sternotomy allowed complete access to the heart for device implantation and heart explantation post-mortem. It also provided access to the LV apex for comprehensive epicardial and epiaortic echocardiography. A customized sewing ring was secured to the LV apex by twelve interrupted sutures and an additional continuous suture. Following CPB initiation and apical coring (28 mm), the prepositioned balloon catheter was externalized through the coring site and positioned into the distal third of the OG. After balloon inflation within the OG, the catheter was partially withdrawn from the right carotid or left femoral artery. Subsequently, the OG was delivered from the apical coring site through the LV across the AV. Echocardiography ensured proper positioning of both the accessory and OG. The accessory was then connected to a centrifugal-flow LVAD (HM3). Following deairing, balloon deflation, and removal, the LVAD was activated, and the animals were gradually weaned from CPB. In instances of hemodynamic stability, an LVAD ramp test was conducted, incrementally increasing pump speed from 4500 to 7200 rpm by 300 rpm every 4 min. Ramp testing was ceased prematurely if complications such as severe arrhythmia or suspected suction events arose.

For euthanasia, the animals received a bolus of potassium chloride (2 mmol/kg IV) intravenously if required. The heart and the LVAD accessory were excised and evaluated for device and OG positioning across the AV. Visible device flow channels were inspected for clot formation, and the AV and mitral valve were examined for any signs of damage.

### 2.4. Statistical Analysis

Metric variables are reported as means with standard deviations (*SD*) and were compared using mean differences (*MD*). Normality was assessed using the Shapiro–Wilk test. Normally distributed variables were tested by paired two-sided Student’s *t*-test, while non-normally distributed variables were assessed using the Wilcoxon signed-rank test. Homoscedasticity was examined using the Levene test. Multiple linear regression was employed to forecast the pump flow rate based on hemodynamic and pump variables, utilizing a best-model approach. A *p*-value < 0.05 was considered significant. All calculations were performed using *R Statistical Software* (version 2021.9.0.351) [31]. Data visualization was conducted using the *ggplot2* package (version 3.4.2) [32]. Multiple linear regression was conducted using the *leaps* package (version 3.1) [33].

## 3. Results

Ten pigs underwent implantation of the second-generation LVAD accessory. The median duration of the experiments, from skin incision to euthanasia following ramp testing, was 6 h and 42 min (interquartile range, 6 h 8 min to 7 h 51 min). Intraoperatively, before device implantation and LVAD support, ventricular fibrillation occurred in 9 out of 10 animals, converted in all cases through intravenous administration of epinephrine and amiodarone and external defibrillation. Blood loss during implantation was mitigated by autologous retransfusion of collected blood through the venous CPB cannula.

### 3.1. Surgical Procedure

Following pericardiotomy, the length of the transventricular OG was determined by measuring the distance between the LV apex and sinotubular junction using echocardiography (Figure 3a). The sewing ring allowed efficient accessory fixation (Figure 3b). The delivery of the OG via a balloon catheter was conducted safely (Figure 3c). Utilizing the HM3 locking mechanism, connecting the pump to the accessory was simple and fast (Figure 3d). Exemplary chest closure was performed in one animal.

### 3.2. Hemodynamic Monitoring

The hemodynamics were initially assessed at the onset of each procedure [34] and continuously throughout device implantation (Table 1). There was an increase in heart rate (101.0 ± 29.4 vs. 126.0 ± 29.2 bpm, *MD* = 25.0 bpm), accompanied by a decrease in mean arterial blood pressure (72.9 ± 14.7 vs. 55.8 ± 25.5 mmHg, *MD* = −17.0 mmHg) and unilateral carotid blood flow (0.49 ± 0.17 vs. 0.30 ± 0.15 L/min, *MD* = −0.19 L/min) from Pre-CPB to LVAD support. With the LVAD providing approximately half of the measured cardiac output during support (5.5 ± 1.2 L/min), the mean LVAD flow rate was 3.1 ± 1.3 L/min, corresponding to a mean rotational speed of 5029 ± 892 rpm. Throughout LVAD support, there were no secondary signs of coronary malperfusion, neither in ECG (no ST-segment elevation) nor in echocardiography (no regional wall motion abnormalities).

Before euthanasia, ramp testing was conducted on eight animals, with four completing all stages. The maximal flow rate achieved was 5.64 L/min at 6900 rpm (Figure 4a). In two animals, the LVAD flow rate was additionally measured using a clamp-on ultrasound sensor positioned behind the HM3 outlet. The estimated flow rate was higher than the one measured, suggesting an overestimation by the HM3 algorithm. Notably, both flow rates revealed a strong correlation (*R* = 0.86, *p* < 0.001) (Figure 4b). Multiple linear regression was employed to predict the estimated and measured pump flow rates based on power, pulsatility index, and rotational speed (adj. *R*^2^ = 0.58 vs. 0.67, *p* < 0.001). After excluding relevant outliers and addressing multicollinearity (e.g., pump power ~ rotational speed, *R* = 0.97, *p* < 0.001) and including hemodynamic factors, the prediction of the measured flow improved significantly, particularly when considering rotational speed, pulsatility index, and mean pulmonary artery pressure (adj. *R*^2^ = 0.69, *p* < 0.001).

### 3.3. Echocardiography

Before and after LVAD implantation, comprehensive epicardial and epiaortic echocardiography was conducted. It confirmed proper OG positioning across the AV in nine cases. Among these, three cases showed the OG was well centered within the AV, four cases between the left and non-coronary cusp, two cases between the right and non-coronary cusp, and one case between the right and left coronary cusp. A reduction in the end-diastolic and end-systolic LV diameter by 16% and 20%, respectively, indicated effective mechanical LV unloading by the LVAD compared to baseline. Mild de novo aortic regurgitation was observed in one case. Further detailed echocardiographic results and evaluations have been described separately [35].

### 3.4. Hematology and Blood Chemistry

Throughout LVAD implantation, free hemoglobin, hemolysis index, and lactate dehydrogenase increased compared to baseline (*p* < 0.001). However, under LVAD support only and compared to CPB, there were no additional changes observed in any of the aforementioned hemolysis markers (*p* > 0.05) (Figure 5a–c), nor of the white blood cell count or hemoglobin (Figure 5d,e). However, there was a slight decrease in platelet count (Figure 5f). Furthermore, under LVAD and compared to baseline, there was an increase in von Willebrand factor (vWF) antigen (Figure 5g). At the same time, vWF activity remained unchanged (Figure 5h), resulting in a decrease in the vWF ratio (Figure 5i).

### 3.5. Post-Mortem Analysis

Visual inspection following heart explantation (Figure 6a) confirmed the echocardiographic finding of OG positioning across the AV in nine animals (Figure 6b). In one case, the OG was positioned below the AV within the LV outflow tract. Across all cases, the visible flow channels of the LVAD, in combination with the accessory, the LV, and the aortic root, showed no signs of clot formation. Moreover, no macroscopic injuries to the AV or mitral valve (Figure 6c,d), and no instances of OG entanglement with subvalvular mitral chords were noted.

## 4. Discussion

The implanted accessory enables LVAD implantation through a single transapical access. This study confirms the device’s anatomic fit and the feasibility of a novel LVAD implantation technique. Using a transventricular OG, the results demonstrate successful redirection of the blood flow through the accessory without signs of thrombus formation or OG entanglement in the papillary muscles or the mitral valve chords. Thereby, the study delivered valuable insights for advancing the accessory and implantation method.

During acute testing and after necropsy, there were no signs of relevant AV dysfunction, coronary malperfusion, or thrombus formation in the aortic root related to the OG across the AV. Future studies should investigate medium- and long-term valve–device interaction, which could cause AV remodeling and dysfunction. Additionally, in silico and in vitro studies using mock circulation loops, comparing different OG configurations and their impact on AV function, coronary perfusion, and thrombogenicity should be performed. To minimize the aforementioned adverse effects, the choice of OG size should consider the patient’s characteristics and AV dimensions. While a larger OG reduces pressure loss, shear stress, and turbulences, minimizing hemolysis and platelet activation, a smaller outflow graft can be inserted more easily, and is likely to affect the AV less. Previous animal studies on experimental LVAD designs reported slight histological changes in the AV. For instance, Tamez et al. tested an intraventricular MVAD with its outflow cannula (approximately 10 mm in diameter) positioned across the AV in both acute and chronic studies using male calves. Thirty days post-implantation, histopathological analysis of the AV showed minimal to mild multifocal subendothelial hyperplasia and minimal neutrophilic inflammation. No impairment of the coronary perfusion was reported [15]. Schima et al. tested an MVAD (HeartWare, Miami Lakes, FL, USA) with a special inflow cannula inserted through the superior pulmonary vein, crossing the left atrium and mitral valve into the LV. In a chronic study, two of eight sheep showed mildly thickened mitral valve leaflets with discrete red discoloration [36]. Referring to the Impella, a microaxial pump for temporary MCS implanted across the AV, the incidence of valvular complications is very low despite more than 300,000 implantations until 2024 [37]. Only a few cases of aortic regurgitation have been reported following percutaneous implantation [38,39,40]. In one case, the repeated back-and-forth movement of the device across the AV and its displacement towards the non-coronary cusp led to histological findings of endothelial denudation and extravasation of red blood cells into the interstitial space without any evidence of neovascularization, suggesting Impella-related mechanical injury [38].

Prospectively, the Impella BTR Early Feasibility Study results will bring new insights into the long-term device–valve interaction [41]. While current Impella pumps are approved for up to 14 days, the Impella BTR is supposed to provide full hemodynamic support for more than one year [42]. Compared to the accessory’s outflow graft, which is 12 mm in diameter, the 7 mm cannula diameter of the Impella BTR is significantly smaller, likely to impair AV function less. Comparing the Impella BTR with conventional LVADs and LVAD with experimental transvalvular outflow grafts, independent of the therapy goal, the Impella BTR provides the least invasive approach but is still in early clinical testing.

In the past, other transventricular LVAD approaches have been proposed, which reveal several disadvantages. Since intraventricular MVADs are not visible from the outside during implantation, additional imaging, such as fluoroscopy, is required to control correct positioning [15]. Rigid outflow designs are not adaptable to the individual angle of the LV outflow tract. Eccentric positioned OGs might lead to additional contact-related AV damage and mechanical stress on the aortic wall. Long-term use could promote AV remodeling, aortic regurgitation, stenosis, and aortic endothelial dysfunction, increasing the risk of ruptures and aneurysm formation. In patients who underwent transcatheter AV replacement before, a rigid outflow design could prove disadvantageous as potential contact between the transaortic outflow and the AV prosthesis might favor damage to either the MVAD or the valve, analogous to the cases reported with Impella [43].

In addition to transvalvular MCS devices, experimental subvalvular devices are under development. One such device is the Flowmaker (FineHeart, Bordeaux, France), an intraventricular, systole-synchronized, wireless device with a subvalvular outflow (pump outlet 1 to 2 cm below AV) [44]. Although the company has received approval from the Czech health authority to start the first-in-human clinical study [45], the long-term stability of its position needs to be investigated, considering reverse myocardial remodeling after device implantation [46].

Prospectively, the tested LVAD accessory will be redesigned to be compatible with other LVADs. The size of the accessory will be adjusted to assess its potential application in pediatric patients and those with small body surface areas. The next-generation LVAD accessory is implanted via lateral thoracotomy—which is theoretically a better approach than sternotomy—without CPB, using a customized introducer sheath and coring tool. Preliminary results from the ongoing acute follow-up trials in sheep, performing LVAD implantation on a beating heart, are promising. In LVAD patients, avoiding median sternotomy eliminates the risks of postoperative sternal dehiscence, deep sternal wound infections, and mediastinitis [47]. Comparing both approaches for sheep requires further study. Irrespective of this, even when using the accessory, there may be cases in which a sternotomy and CPB are necessary or unavoidable in case of concomitant cardiac surgery procedures, interventricular septal defects, or LV thrombi.

To address two main limitations of the current study—small sample size and a healthy porcine model—future investigations will involve larger animal cohorts featuring an appropriate HF model characterized by reduced ejection fraction [48]. It is generally advisable to test variations in device sizing across different porcine age groups, assessing anatomical compatibility for pediatric and adult patients. An oversized transvalvular OG may lead to AV dysfunction and coronary malperfusion due to its excessively large diameter, interfering with proper AV closure and blood flow to the coronary arteries. Forthcoming chronic experiments should encompass comprehensive histopathological assessments of the heart and all relevant end organs. The histological examination of the AV is critical, evaluating the potential impact of medium- and long-term valve–device interaction, which may contribute to valvular remodeling.

## 5. Conclusions

This feasibility study on a second-generation LVAD accessory demonstrates its anatomic fit and proves a novel implantation technique for successfully redirecting the LVAD blood flow through the LV. Future studies will investigate minimally invasive implantation without CPB and long-term application in sheep, focusing on assessing AV function and device biocompatibility.

## Figures and Tables

**Figure 2 bioengineering-11-00848-f002:**
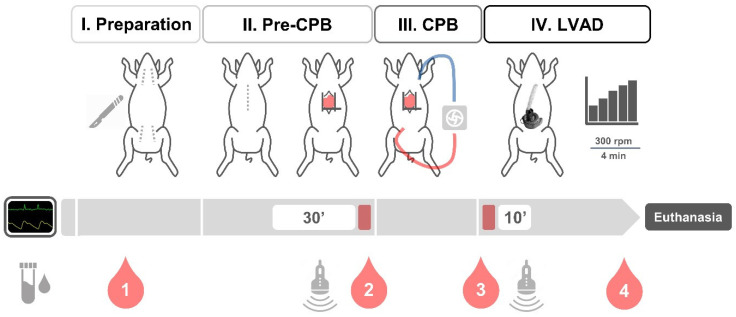
Study plan for acute animal testing. The surgical procedure comprised four phases: (I) Preparation after skin incision, (II) before cardiopulmonary bypass (pre-CPB) after sternotomy, (III) CPB, and (IV) left ventricular assist device (LVAD) support with ramp test. Hemodynamics were monitored continuously and compared before and during LVAD support, applying periods of 30 and 10 min. Blood samples were collected (1) after arterial catheterization (baseline), (2) before CPB, (3) before, and (4) after LVAD support (before euthanasia).

**Figure 3 bioengineering-11-00848-f003:**
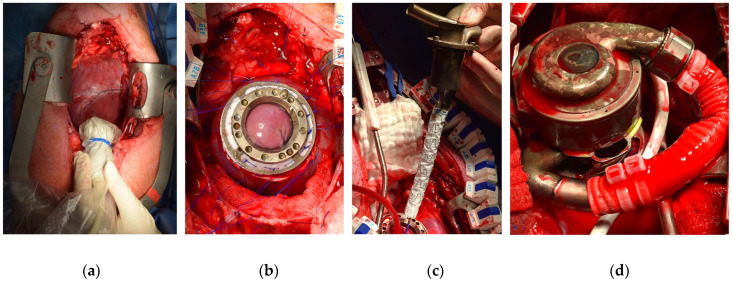
Implantation of the left ventricular assist device with accessory requiring single transapical access: (**a**) Epicardial echocardiography to assess aortic valve function prior to device implantation. (**b**) Customized sewing ring on left ventricular apex. (**c**) Insertion of the accessory comprising a stent graft as an outflow using a retrogradely positioned balloon catheter placed in the distal third of the outflow graft. (**d**) Left ventricular assist device connected to the accessory.

**Figure 4 bioengineering-11-00848-f004:**
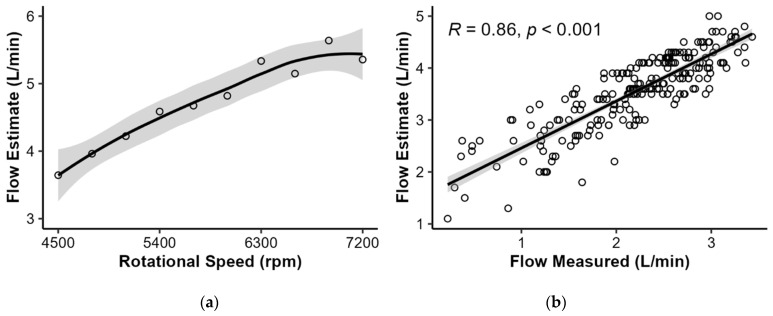
Estimated flow rate during ramp testing and comparison of measured and estimated flow in two animals: (**a**) Estimated flow rate during ramp test. (**b**) Correlation between estimated and measured pump flow rate (*R* = 0.86, *p* < 0.001).

**Figure 5 bioengineering-11-00848-f005:**
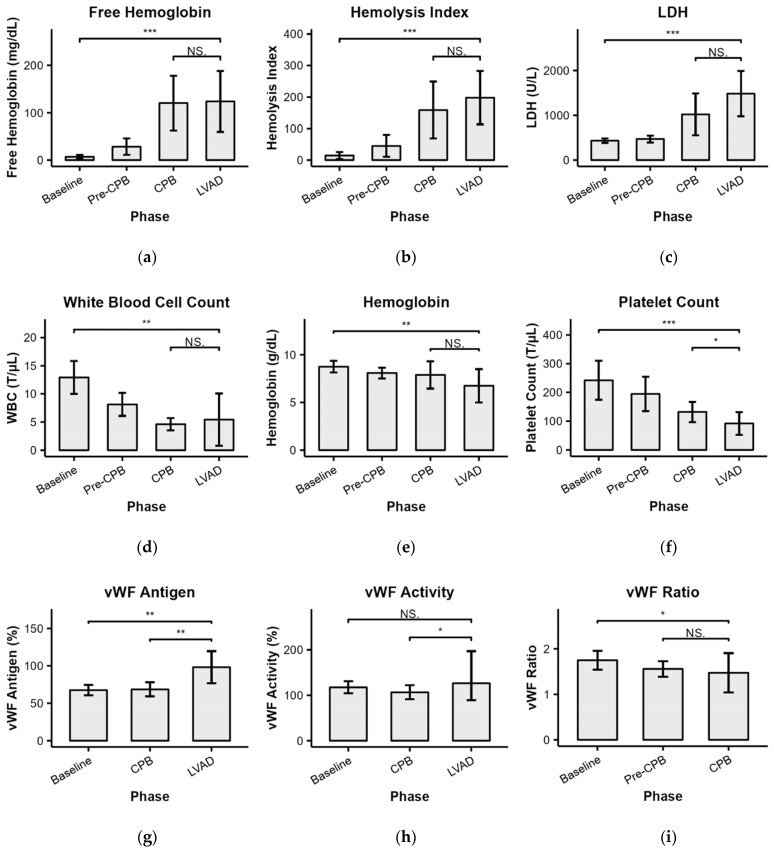
Intraoperative hemolysis markers, blood cell counts, and von Willebrand factor (vWF): (**a**) Plasma-free hemoglobin. (**b**) Hemolysis index. (**c**) Lactate dehydrogenase (LDH). (**d**) White blood cell count (WBC). (**e**) Hemoglobin. (**f**) Platelet count. (**g**) vWF antigen. (**h**) vWF activity. (**i**) vWF ratio. NS, non-significant; * *p* < 0.05, ** *p* < 0.01, *** *p* < 0.001.

**Figure 6 bioengineering-11-00848-f006:**
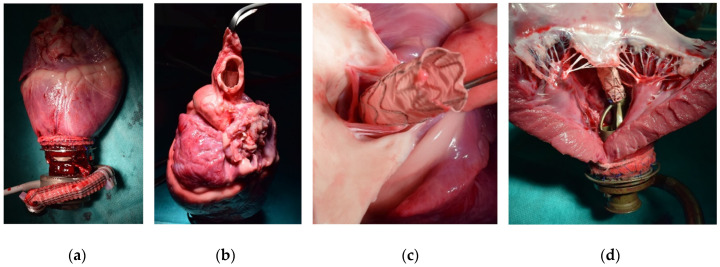
Visual inspection of the heart during necropsy: (**a**) Pump with accessory at the heart apex. (**b**) Tip of outflow graft in ascending aorta. (**c**) Outflow graft across the aortic valve. (**d**) Accessory inflow and proximal OG half within the left ventricle.

**Table 1 bioengineering-11-00848-t001:** Hemodynamics, vital parameters, carotid blood flow, and pump parameters during LVAD implantation.

	Pre-CPB	LVAD	*MD*	*p*
HR (bpm)	101.0 ± 29.4	126.0 ± 29.2	25.0	0.032
SBP (mmHg)	98.9 ± 19.8	96.4 ± 35.2	−2.5	0.940
DBP (mmHg)	59.7 ± 15.8	35.0 ± 27.7	−24.7	0.018
MBP (mmHg)	72.9 ± 14.7	55.8 ± 25.5	−17.0	0.045
sPAP (mmHg)	25.9 ± 5.6	31.0 ± 13.0	5.2	0.097
dPAP (mmHg)	14.3 ± 3.9	20.1 ± 8.8	5.8	0.063
mPAP (mmHg)	19.4 ± 4.1	25.4 ± 11.1	6.0	0.080
CVP (mmHg)	9.4 ± 3.4	9.9 ± 4.3	0.5	0.321
CO (L/min)	7.9 ± 2.3	5.5 ± 1.2	−2.4	0.120
BT (°C)	36.6 ± 1.1	35.3 ± 0.8	−1.2	0.008
CCA flow (L/min)	0.49 ± 0.17	0.30 ± 0.15	−0.19	0.016
Flow (L/min)		3.11 ± 1.28		
Speed (rpm)		5029 ± 892		

BT, body temperature; CCA, common carotid artery; CO, cardiac output; CVP, central venous pressure; DBP, diastolic blood pressure; HR, heart rate; MBP, mean blood pressure; MD, mean difference; PAP, pulmonary arterial pressure (s, systolic; d, diastolic; m, mean); SBP, systolic blood pressure.

## Data Availability

The data supporting this article’s findings are available from the corresponding author, Wolfgang Bothe [wolfgang.bothe@uniklinik-freiburg.de], upon reasonable request.

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
