# Peer review of "In Vivo Testing of a Second-Generation Prototype Accessory for Single Transapical Left Ventricular Assist Device Implantation"

_bioengineering, 2024, doi:10.3390/bioengineering11080848_

Round 1

Reviewer 1 Report

Comments and Suggestions for Authors

Authors should be advise there are some instances where sentence structure could be revised for smoother readability.

Study is good, particularly as it is directly in a human analogue, could be expanded to more animals ,but it would increase greatly the cost. Albeit, would be great to have different aged animals for anatomical and physiological considerations.

any noteworthy complications? 

how long did the study last? this is relevant to comprehand the complications that will arise

Would also be good to have histological staining to see remodeling or fibrosis

Comments on the Quality of English Language

Well written, sure a few things here and there but nothing major

Author Response

Comment 1.1: Authors should be advice there are some instances where sentence structure could be revised for smoother readability.

Response 1.1: Thank you for your valuable feedback. We have carefully reviewed and revised the manuscript to enhance sentence structure and improve overall readability. We hope these changes make the content more accessible and clear for the readers.

Comment 1.2: Study is good, particularly as it is directly in a human analogue, could be expanded to more animals, but it would increase greatly the cost. Albeit, would be great to have different aged animals for anatomical and physiological considerations.

Response 1.2: We agree that expanding the study to include more animals would enhance the power. In our current study, we selected pigs based on their weight to ensure an appropriate heart size for our device. Smaller pigs tend to have hearts that are too small for accurate testing, while larger pigs pose significant challenges in perioperative handling. Although we recognize the benefits of using animals of different ages for anatomical and physiological considerations, our selection criteria were primarily focused on achieving a balance between practicality and scientific validity. We acknowledge that incorporating a broader age range would provide valuable data on the device's performance across different physiological stages. Future studies could certainly benefit from including a wider variety of animal subjects. However, as you mentioned, the increased cost and logistical complexity are considerable factors. We appreciate the suggestions and will consider them for future studies. We added these aspects to the discussion of this study. Please see the following lines: 351–355

Point 1.3: How long did the study last? This is relevant to comprehend the complications that will arise.

Response 1.3: Thank you for your comment. This was an acute study with euthanasia performed after LVAD ramp testing. Information on the duration of the surgery has been added to the results. We understand the importance of assessing long-term complications and are currently preparing a study to evaluate the device over a 30-day period in a large animal model. This study will provide valuable insights into the long-term performance and potential complications. Please see the following lines: 205–211

Point 1.4: Would also be good to have histological staining to see remodeling or fibrosis.

Response 1.4: We agree that histological staining would provide valuable insights into cardiac remodeling and fibrosis. However, the duration of this study needed to be longer to effectively evaluate these processes. We recognize the importance of this aspect and plan to conduct comprehensive histological assessments in our future long-term studies. These assessments will include the histology of the heart and all end organs to evaluate any potential changes or complications thoroughly. Please see the following lines: 255–259

Reviewer 2 Report

Comments and Suggestions for Authors

Meissner et al. describe a procedure for the implantation of a left ventricular assist device with a rather minimally invasive approach. The lateral thoracotomy is essential here, as it is expected to reduce adhesions for a potential later transplantation. A minimally invasive approach is to be preferred for the expected patient clientele. Patients who require mechanical circulatory support are already at a higher risk of multiple organ failure. The potential renunciation of the use of the heart-lung machine initially appears to make sense here. However, it must first be ensured that there is no septal defect and no LV thrombi are present. Otherwise, the use of a heart-lung machine makes more sense, even if it puts additional stress on the body. These circumstances should be mentioned in the discussion.

Author Response

Point 2.1: Meissner et al. describe a procedure for the implantation of a left ventricular assist device with a rather minimally invasive approach. The lateral thoracotomy is essential here, as it is expected to reduce adhesions for a potential later transplantation. A minimally invasive approach is to be preferred for the expected patient clientele. Patients who require mechanical circulatory support are already at a higher risk of multiple organ failure.

Response 2.1: We agree that a minimally invasive approach, such as the lateral thoracotomy, is advantageous for patients requiring mechanical circulatory support. This technique not only minimizes adhesions, which is crucial for potential future transplantation but also aligns with the goal of reducing overall surgical trauma. Given that these patients are already at a higher risk of multiple organ failure, the benefits of a less invasive procedure are significant.

Point 2.2: The potential renunciation of the use of the heart-lung machine initially appears to make sense here. However, it must first be ensured that there is no septal defect and no LV thrombi are present. Otherwise, the use of a heart-lung machine makes more sense, even if it puts additional stress on the body. These circumstances should be mentioned in the discussion.

Response 2.2: We agree that using a heart-lung machine might be beneficial under certain circumstances, such as in the presence of a septal defect or LV thrombi. However, avoiding the heart-lung machine has several significant advantages. Off-pump implantations reduce the risk of systemic inflammatory response, minimize bleeding and the need for blood transfusions, shorten recovery times, decrease the incidence of neurological complications, and lower the risk of renal dysfunction. In the discussion, we have included the circumstances under which a heart-lung machine might be necessary. Please see the following lines: 244–250
